# ENSAM: an efficient foundation model for interactive segmentation of 3D medical images

Elias Stenhede[1,2][0009−0005−2654−4553],
Agnar Martin Bjørnstad[1,2][0009−0005−4207−6278], and
Arian Ranjbar[1][0000−0002−0422−2255]

[1] Medical Technology & E-health, Akershus University Hospital,
Lørenskog, Norway
[2] Faculty of Medicine, University of Oslo, Oslo, Norway
arian.ranjbar@medisin.uio.no

**Abstract.** We present ENSAM (Equivariant, Normalized, Segment Anything Model), a lightweight and promptable model for universal 3D medical image segmentation. ENSAM combines a SegResNet-based encoder with a prompt encoder and mask decoder in a U-Net-style architecture, using latent cross-attention, relative positional encoding, normalized attention, and the Muon optimizer for training. ENSAM is designed to achieve good performance under limited data and computational budgets, and is trained from scratch on under 5,000 volumes from multiple modalities (CT, MRI, PET, ultrasound, microscopy) on a single 32 GB GPU in 6 hours. As part of the CVPR 2025 Foundation Models for Interactive 3D Biomedical Image Segmentation Challenge, ENSAM was evaluated on the hidden test set with multimodal 3D medical images, obtaining a DSC AUC of 2.404, NSD AUC of 2.266, final DSC of 0.627, and final NSD of 0.597, outperforming two previously published baseline models (VISTA3D, SAM-Med3D) and matching the third (SegVol), surpassing its performance in final DSC but trailing behind in the other three metrics. In the coreset track of the challenge, ENSAM ranks 5th of 10 overall and best among the approaches not utilizing pretrained weights. Ablation studies confirm that our use of relative positional encodings and the Muon optimizer each substantially speed up convergence and improve segmentation quality.

**Keywords:** Medical Imaging · Multimodal · Interactive Segmentation

## 1 Introduction

Accurate segmentation of three-dimensional (3D) or two-dimensional plus time (2D+t) medical images has become fundamental for numerous clinical tasks, including diagnosis, treatment planning, and disease monitoring. While 3D images provide much richer spatial context compared to 2D images, technical challenges arise in processing and storing large amounts of high-resolution 3D data. In the last decade, specialized deep learning models have shown success in automating segmentation tasks when trained using high-quality pixel-level labels [22].

More recently, the advent of large pretrained foundation models in natural language processing has demonstrated that models trained on massive and diverse datasets can generalize effectively to downstream tasks [7, 36, 6, 39], surpassing the performance of specialized models.

Motivated by this, foundation models for natural image segmentation have been developed, most notably SAM [20] for 2D and subsequently SAM2 [37] for 2D+t. While providing impressive segmentations for natural images, they do not immediately provide useful segmentations when applied to medical images. To address this performance gap, the 2025 CVPR Workshop on Foundation Models for Medical Vision was established, including a challenge aimed at improving segmentation accuracy on medical modalities. In this paper, we present our contribution to that effort: an efficient SAM model for medical 3D imaging.

## 1.1   Related work

Several previous attempts at building a SAM for medical imaging exist. Med-SAM [26] adapts SAM to the medical domain by fine-tuning on medical segmentation datasets, achieving improved performance across several imaging modalities, but is limited by only supporting an initial bounding box and no subsequent clicks. It further only supports 2D slices and lacks volumetric consistency. Med-SAM2 [28] adapts SAM2 for and supports segmentation of 2D+t and 3D medical images, but also lacks support for iterative segmentation refinement.

Models inspired by but not directly utilizing weights from SAM or SAM2 have also been proposed. SAM-Med3D [40] demonstrates the feasibility of training using medical data only and supports iterative refinement, but fails to generalize well to unseen classes and requires many interactions to match the performance of task-specific baselines such as nnU-Net [13]. SegVol [8] introduces more prompt types, including text, boxes, and clicks, and is trained on Computed Tomography (CT) images only.

Other models build on established segmentation architectures to enhance performance in medical imaging tasks. SegResNet [29], a U-Net-based architecture augmented with a variational decoder head for regularization, has proven highly effective, winning multiple 3D medical image segmentation challenges [32, 31, 30]. It serves as the backbone in VISTA3D [11], a model trained on both CT and MRI data, which leverages supervoxels generated by SAM during training in addition to conventional segmentation masks.

The model nnInteractive [9] advances the state-of-the-art further by supporting bounding box, click, lasso, and scribble prompts, with the latter two providing stronger guidance. It is trained on a multimodal set of 3D medical datasets using both traditional segmentation masks and supervoxels from SAM and SAM2. Unlike the other models that rely on cross-attention, nnInteractive integrates user input through dense feature maps.

### 1.2   Objective and contribution

As the field of interactive 3D medical image segmentation rapidly evolves, each new model often demonstrates improvements over selected baselines. However, a unified and independent comparison of these models on a common benchmark has not yet been established. This challenge addresses that gap by evaluating participants on a standardized hidden test set comprising multiple imaging modalities, enabling a fair and unbiased comparison of methods based solely on their generalization and segmentation performance.

ENSAM (Equivariant, Normalized, SAM in 3D) is designed to improve both training efficiency and segmentation accuracy, addressing the often prohibitive computational cost of training foundation models. Our objective is to achieve performance comparable to, or surpassing, current state-of-the-art models, while training on a single GPU. The name ENSAM also subtly reflects this goal, as "ensam" (Swedish spelling) means sole/lonely in Germanic languages.

Our proposed model is inspired by SAM and the SegResNet architecture, and introduces several improvements, such as relative position encodings in 3D together with a normalized attention mechanism for the encoded user input, trained by a Muon optimizer in place of Adam. All modifications are aimed at accelerating training and enhancing data efficiency, while being scalable to larger setups. Furthermore, our solution targets the coreset challenge, a sub-track of the challenge, limiting the training dataset to 10% of the full set.

## 2   Method

The goal of the challenge is to develop a foundation model for universal medical image segmentation; that is, given a medical image, the model should segment any anatomical or pathological structure indicated by a user prompt (specifically, a bounding box or point-based clicks). Evaluation is conducted through a simulated interactive setting, in which the model receives an image during inference, with or without an initial bounding box, followed by five simulated "clicks" representing iterative corrections made by a clinician.

To address this task, we propose a model based on three components: an image encoder, a prompt encoder, and a mask decoder, configured into a U-Net architecture. User prompts are integrated into the model via cross-attention mechanisms applied at the bottleneck layer. With the chosen structure, we are capable of simultaneously training all components, end-to-end. The following subsections detail the proposed method, and an overview of the architecture is provided in Figure 1.

### 2.1   Image Encoder

With the advent of vision transformers, several efforts have been made to adapt transformer-based backbones for use as image encoders in segmentation models. However, most benchmarks continue to show that CNN-based encoders outperform their transformer-based counterparts [3]. We adopt an architecture based

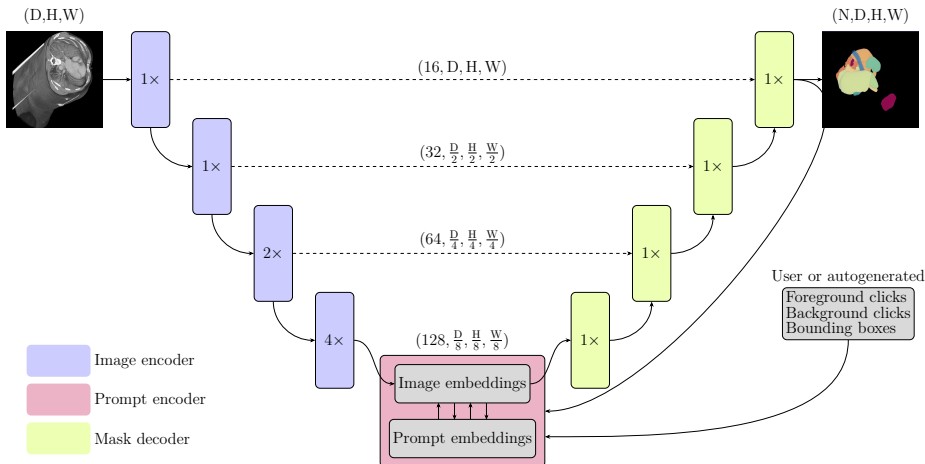

Fig. 1: Network architecture of the proposed interactive segmentation model, consisting of three main components: image encoder, prompt encoder, and mask decoder. During an interactive segmentation session on a single sample, the image encoder runs once, while the decoder updates the segmentation with each new user input.

on the SegResNet model, [29], which comprises a cascade of residual blocks interleaved with downsampling layers. An overview of the image encoder is illustrated in Figure 2.

## 2.2   Prompt Encoder

The prompt encoder is responsible for taking input from the user and encoding it in a format compatible with the rest of the model. The prompt encoder currently supports 3D bounding boxes as well as foreground and background clicks. All user interactions are represented as unit vectors with the same dimension $d$ as the image embeddings at the bottom latent space of the U-Net. In its current implementation, boxes are represented by a pair of unit vectors and clicks by a single vector, one for foreground and another for background clicks.

Each prompt and image embedding is associated with a 3D coordinate. This coordinate information is essential for the mask decoder to reason about spatial relations between prompts and image content. Traditionally, absolute positional information has been added element-wise to embedding vectors [20], [26]. However, absolute encoding breaks equivariance. Using methods that instead encode relative positional information has been shown to improve training efficiency and final model performance both in 1D [38] as well as 2D and 3D tasks [34].

**Lie Rotational Positional Encoding.** To include relative positional information when computing attention between embedding vectors, the attention blocks

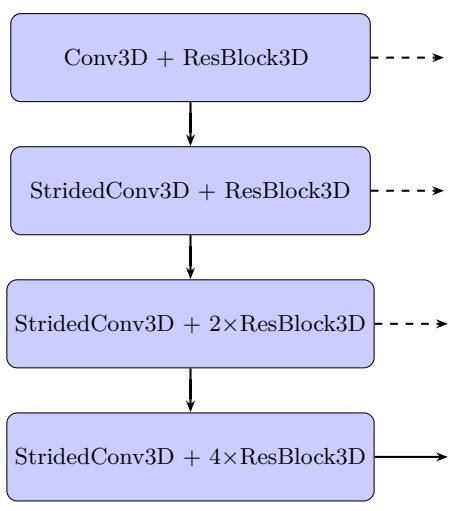

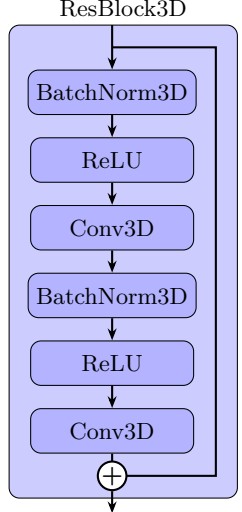

(a) The image encoder consists of four blocks. The initial Conv3D layer transforms the image from single-channel to 16 channels. The StridedConv3D layers reduce the spatial dimensions by a factor of 2 and double the number of channels.

(b) Each residual block contains convolutional layers with skip connections. Through all layers, the channel dimension is kept constant, allowing for element-wise addition of the residual activations.

Fig. 2: Architecture overview: (a) image encoder; (b) residual block used within the encoder.

are given pairs of embedding vectors and coordinates. Technically, positional information is encoded by applying position-dependent rotation matrices to key and query vectors. As previously noted in [34], Lie algebras provide a suitable framework in this setting, as the group of rotations SO(n) can be generated from the Lie algebra $\mathfrak{so}(n)$. In other words, for an embedding vector $e_i$ with coordinate $p_i = (x_i, y_i, z_i)$, the corresponding rotation matrix can be written as

$$R(p_i) = \exp\left(A_x x_i + A_y y_i + A_z z_i\right), \tag{1}$$

where $A_x$, $A_y$, $A_z$ are learnable, skew-symmetric matrices of size $d \times d$, where $d$ is the embedding dimension. Each matrix is parameterized by $d(d-1)/2$ values due to the skew-symmetry ($A^\top = -A$).

Letting $q_i$ and $k_j$ be key and query vectors with positions $p_i = (x_i, y_i, z_i)$ and $p_j = (x_j, y_j, z_j)$ respectively. The attention scores between $q_i$ and $k_j$ are then

calculated as

$$
\begin{aligned}
\text{AttnScore}(q_i, k_j) &= (R(p_i)q_i)^\top (R(p_j)k_j) \\
&= q_i^\top R(p_i)^\top R(p_j)k_j \\
&= q_i^\top \exp\left(A_x x_i + A_y y_i + A_z z_i\right)^\top \exp\left(A_x x_j + A_y y_j + A_z z_j\right) k_j \\
&= q_i^\top \exp\left(-A_x x_i - A_y y_i - A_z z_i\right) \exp\left(A_x x_j + A_y y_j + A_z z_j\right) k_j \\
&= q_i^\top \exp\left(A_x(x_j - x_i) + A_y(y_j - y_i) + A_z(z_j - z_i)\right) k_j \\
&= q_i^\top R(p_j - p_i)k_j,
\end{aligned}
$$

which shows that the attention scores depend only on the relative position $p_j - p_i$ and coincides with the standard attention calculation when $p_j = p_i$, as $\exp(0) = I$.

**Normalized Attention.** Recent work by Loshchilov et al. [24] introduces a normalized transformer architecture, which can converge in 4-20 times fewer training steps compared to the standard transformer, primarily demonstrated on 1D natural language tasks. In ENSAM, we extend this approach to the 3D medical image domain, combining it with LieRE. We hypothesize that their benefits, namely faster convergence and improved numerical stability, can generalize to volumetric data.

The normalized transformer replaces traditional layer normalization (e.g., RMSNorm [42] or LayerNorm [2]) and weight decay with $\ell_2$ normalization applied to all weight matrices after each optimization step. An additional $\ell_2$ normalization of activations is also performed. As a result, attention and MLP outputs are constrained to lie on a unit hypersphere, which requires a modified residual update strategy. Specifically, the standard residual addition:

$$
x \leftarrow x + \text{Block}(x) \tag{2}
$$

is replaced by

$$
x \leftarrow \text{Norm}\left(\text{Norm}(x) + \lambda(\text{Norm}(\text{nBlock}(x)) - \text{Norm}(x))\right). \tag{3}
$$

In eq. (3), Norm denotes $\ell_2$ normalization and $\lambda \in \mathbb{R}^d_+$ is an eigen learning rate that is learned for each block in the model. Block and nBlock denote the blocks in a transformer architecture and a normalized transformer architecture, respectively. Steps performed by cross-attention and MLP layers are performed using the same logic. This update rule can be interpreted as a constrained optimization step on the unit hypersphere, which empirically stabilizes training and accelerates convergence.

For initialization of e.g. $\lambda$, we used the values recommended by the original authors. For such implementation, specific details and theoretical justifications, we refer readers to the original study [24].

**Image-Prompt Interaction.** The interaction between user prompts and image embeddings builds on the original SAM model, with modifications to the attention mechanism, positional encoding, and postprocessing. Besides the image embeddings, the prompt encoder processes user inputs and, when available, segmentation logits from the previous step. These segmentation logits are down-sampled using strided convolutions to align with the image embeddings and are added element-wise. The interaction between user input and the modified image embeddings follows a four-step process, using normalized attention and relative positional encoding as core components.

1. Normalized self-attention is applied to the prompt embeddings.
2. The prompt embeddings attend to the image embeddings.
3. The updated prompt embeddings are passed through a multi-layer percep-tron (MLP) with ReLU activation and a hidden dimension of $2d$.
4. The image embeddings attend to the updated prompt embeddings.

All four steps incorporate residual connections on the hypersphere using Equation (3). This four-step process is repeated twice and is illustrated in Figure 3.

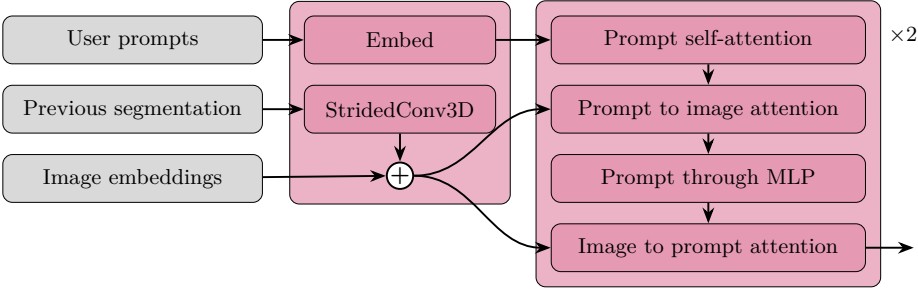

Fig. 3: The prompt encoder module encodes user input and modifies the image embeddings before it is passed to the mask decoder. If a previous segmentation exists, it is incorporated into the image embeddings via strided convolutions, allowing for iterative refinement.

## 2.3   Mask Decoder

The mask decoder mirrors the image encoder, except for single residual blocks per upsampling layer. The activations from skip-connections are concatenated along the channel dimension and processed by a single ResBlock3D, followed by trilinear upsampling. The final layer outputs logits with the same shape as the input, containing the segmentation mask.

### 2.4   Model Training and interaction simulation

During training, user interactions are simulated. If provided, initial prompts are given as bounding boxes, calculated using the ground truth labels with an added random offset, to mimic human generation. An iterative refinement click is then placed in the middle of the largest error region. In case the largest error region is an undersegmentation, a foreground click is placed; otherwise, a background click is used. In total, a bounding box and five clicks are provided per training step.

To provide supervision for the model, we use the sum of generalized dice loss and cross-entropy, as compound loss functions have been proven to be robust in various medical image segmentation tasks [25]. Specifically, the cross-entropy carries double the weight,

$$\mathcal{L} = \mathcal{L}_{\text{Dice}} + 2 \cdot \mathcal{L}_{\text{CE}} = 1 - \frac{2\sum_i p_i g_i}{\sum_i p_i^2 + \sum_i g_i^2} - \frac{2}{N}\sum_i g_i \log p_i, \tag{4}$$

where $N$ is the number of voxels, $p$ denotes the prediction and $g$ the ground-truth. The loss is averaged across all iterative steps.

To train the model in as high a resolution as possible without exceeding the GPU memory constraints, gradient accumulation and a batch size of one are used. The usage of a batch size of one is also partly motivated by the varying data shapes. A standard torch dataset/dataloader setup was used with 32 worker threads for parallel data loading. Data preprocessing and augmentation were performed on the fly within the dataset, as it did not serve as a bottleneck for the training pipeline.

**Muon optimizer** Following its recent success in speedrunning training of image classification and language models, [17, 23], we investigate if the Muon optimizer [18] is effective for segmentation models. The Muon optimizer has, to the best of our knowledge, not been benchmarked for segmentation tasks at the time of writing. Unlike traditional optimizers like Adam or SGD, Muon operates on 2-dimensional weight matrices. To apply Muon to ENSAM, all weights with dimension $\geq 2$ are flattened beyond the first dimension. For example, 3D convolutional kernels are 5-dimensional and need flattening. For parameters with dimension 1, Adam is used as per usual.

Muon replaces the conventional gradient descent update with a step along $UV^\top$, where $U\Sigma V^\top$ is the singular value decomposition (SVD) of the gradient matrix. Rather than computing the full SVD, Muon employs an efficient approximation [4, 5], which has been shown to achieve similar performance [23] while significantly reducing computational cost.

### 2.5   Coreset selection strategy

The coreset track in the challenge required the use of no more than 10% of the total training data, corresponding to a maximum of 4,471 samples. To ensure

diverse representation under this constraint, we aimed to select an approximately equal number of samples from each dataset. In cases where a dataset contained fewer than $4471/N$ samples (where N is the number of eligible datasets), all available samples from that dataset were included.

Some datasets were excluded from the coreset selection process. The CT Aorta dataset was omitted due to apparent issues with image normalization. In addition, the microscopy datasets were excluded as many of them had issues in the provided format. Since the ground truth annotations were stored using the `uint8` format, this led to instance merging due to label value overflow.

### 2.6   Post-processing

Although the model is trained to segment one instance at a time, multiple instance prompts are typically provided during inference. To handle this, only one encoder pass is needed for the input, while the prompt encoder and decoder can be run in parallel for each instance. The final segmentation assigns each voxel to the instance with the highest output logit, or to the background if no logit exceeds a predefined threshold, such as 0.

To ensure that each instance is assigned at least one voxel, the logits are adjusted by adding a constant inside the bounding box of any instance that initially has no assigned voxels. If every instance already has at least one voxel assigned, no changes are made to the logits.

## 3   Experiments

### 3.1   Dataset and evaluation metrics

The development set is compiled by the organizers of the CVPR 2025 Foundation Models for Interactive 3D Biomedical Image Segmentation Challenge. This includes data normalization. The development set is an extension of the CVPR 2024 MedSAM on Laptop Challenge [27], including more 3D cases from public datasets [3] and covering commonly used 3D modalities including CT, MRI, Positron Emission Tomography (PET), Ultrasound (US), and microscopy images. The hidden test set is created by a community effort where all the cases are unpublished. The annotations are either provided by the data contributors or annotated by the challenge organizer with 3D Slicer [19] and MedSAM2 [28]. In addition to using all training cases, the challenge contains a coreset track, where participants can select 10% of the total training cases for model development. The solution proposed in this paper specifically targets the latter coreset track.

Each interactive segmentation is evaluated using the Dice Similarity Coefficient (DSC) and Normalized Surface Distance (NSD), which measure the overlap of segmentation regions and the accuracy of boundaries, respectively.

---

[3] A complete list is available at https://medsam-datasetlist.github.io/

Ranking of participants is performed using four metrics: Area Under the Curve (AUC) for DSC and NSD, as well as final AUC and NSD. Denoting $\text{AUC}_i$ as the DSC after the $i$-th user interaction, the AUC is calculated as

$$\text{AUC\_DSC} = \frac{1}{2}\left(\text{DSC}_1 + 2\cdot\text{DSC}_2 + 2\cdot\text{DSC}_3 + 2\cdot\text{DSC}_4 + \text{DSC}_5\right). \quad (5)$$

The initial bounding box prediction is excluded from this calculation, as it is optional. The same formula is used for computing AUC\_NSD, mutatis mutandis. The four metrics intend to capture both the iterative refinement and final predictions.

Finally, to ensure practical applicability, inference time is capped at 90 seconds per class. Any submission exceeding this limit receives a score of zero for both DSC and NSD on the corresponding test case.

### 3.2   Implementation details

**Preprocessing** Following the practice in MedSAM [26], all images were preprocessed by the challenge organizers into `.npz` format with an intensity range normalized to $[0, 255]$. For CT images, the Hounsfield units were normalized using standard window width and level settings: soft tissue (W:400, L:40), lung (W:1500, L:-160), brain (W:80, L:40), and bone (W:1800, L:400). Subsequently, the intensity values were rescaled to the range of $[0, 255]$. For other images, the intensity values were clipped to the range between the 0.5th and 99.5th percentiles before rescaling them to the range of $[0, 255]$. If the original intensity range is already in $[0, 255]$, no preprocessing was applied.

**Environment settings** The development environments and requirements are presented in Table 1.

Table 1: Development environments and hardware.

| Component | Specification |
| --- | --- |
| System | Debian 12 |
| CPU | Intel(R) Core(TM) i9-14900KF |
| RAM | 2×48 GB; 4800 MT/s |
| GPU | NVIDIA GeForce RTX 5090 32 GB |
| CUDA version | 12.8 |
| Programming language | Python 3.12 |
| Deep learning framework | PyTorch 2.7.0, Torchvision 0.22.0 |

**Training protocols** Training was performed after coreset selection, which involved nearly uniform sampling across datasets. Therefore, oversampling was not

employed. Each training epoch consisted of sampling every data instance once in randomized order and generating boxes or clicks for one randomly selected labelled instance from the label data.

Part of the datasets used during training included irrelevant regions surrounding the areas of interest. To focus computational resources on relevant structures, training volumes were randomly cropped around the labelled regions with a variable margin of 1 to 64 voxels in each spatial dimension. After cropping, volumes that exceeded a predefined size threshold were downscaled via max pooling to fit within GPU memory constraints. The shapes of training volumes varied across samples. However, to ensure compatibility with the network architecture, all spatial dimensions were adjusted to be divisible by 8 by zero padding.

Following the spatial augmentations, the volumes were converted from `uint8` format to a range between $[0, 1]$, and an intensity augmentation was applied with a probability of 0.5. Specifically, one of the following was randomly applied: bias field distortion, Gaussian smoothing, or histogram shift.

Table 2: Parameters used during model training. FLOPs were calculated for one forward pass with the maximum patch volume and only one user interaction.

| Parameter | Value |
|---|---|
| Batch size | 1 |
| Gradient accumulation steps | 4 |
| Patch size | Variable |
| Maximum patch volume | $4{,}194{,}304 \approx 161^3$ |
| Simulated clicks per step | 5 |
| Total epochs | 15 |
| Optimizer | Muon and AdamW |
| Muon momentum | 0.95 |
| Initial learning rate | $10^{-3}$ |
| Learning rate scheduler | Halved at epochs 2, 5, 10 |
| Training time | 6 hours |
| Loss function | Soft Dice + 2·BCE |
| Number of model parameters | 5.5 M |
| Number of FLOPs | 368 G |

## 3.3   Ablations

To evaluate the contributions of LieRE and Muon, we conducted an ablation study. First, ENSAM was trained using absolute position encoding with the Adam optimizer. Next, we replaced absolute encoding with LieRE while retaining the Adam optimizer. Finally, ENSAM was trained using both LieRE and the Muon optimizer. The results can be found in Figure 4, and we note that relative

position encoding and switching optimizer improve training speed, making the model fit faster to the training data.

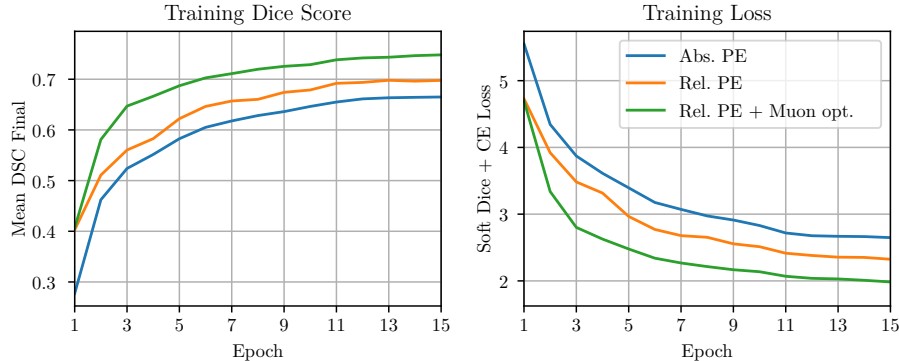

Fig. 4: Three variants of ENSAM trained on the same coreset using the same seed. Relative position encodings improve training efficiency over absolute position encodings. The Muon optimizer further improves upon the relative position encodings. Besides speeding up training, the model trained with Muon also ends up at a better final loss.

## 4 Results and discussion

### 4.1 Quantitative results on validation set

Our proposed model is benchmarked against four previously published interactive segmentation models across all five modalities, and the results are shown in Table 3. On all five modalities, either VISTA3D or SegVol obtains the highest score. Among the five modalities, ENSAM is second in ultrasound, third in MRI and microscopy, and fourth in CT.

### 4.2 Fair comparison and reporting standards

Common pitfalls in evaluating segmentation models include confounding performance boosters, lack of well-configured baselines, insufficient testing data, and inconsistent use of evaluation metrics [14]. In this work, the same evaluation data and metrics are used across all methods, providing a transparent and accurate depiction of each model's performance. That being said, our approach is trained from scratch, only using 10% of the full challenge dataset, without relying on pretrained model weights. Further, our method is trained on a single GPU with 32 GB of VRAM for 6 hours as opposed to the baseline methods that were originally trained using 100-1000 times more computational resources. Lastly,

Table 3: Quantitative evaluation results on the validation set for the coreset track. The maximum value for DSC AUC and NSD AUC is 4, while the maximum value for DSC Final and NSD Final is 1.

| Modality | Method | DSC AUC | NSD AUC | DSC Final | NSD Final |
|---|---|---|---|---|---|
| CT | SegVol | 2.98 | 3.12 | 0.75 | 0.78 |
| | ENSAM (ours) | 2.03 | 1.90 | 0.50 | 0.47 |
| | VISTA3D | 2.81 | 2.84 | 0.72 | 0.73 |
| | SAM-Med3D | 2.28 | 2.27 | 0.57 | 0.57 |
| MRI | SegVol | 2.67 | 3.15 | 0.67 | 0.79 |
| | ENSAM (ours) | 1.84 | 2.07 | 0.45 | 0.51 |
| | VISTA3D | 2.53 | 2.82 | 0.65 | 0.73 |
| | SAM-Med3D | 1.76 | 1.81 | 0.45 | 0.46 |
| Microscopy | SegVol | 2.04 | 3.47 | 0.51 | 0.87 |
| | ENSAM (ours) | 1.27 | 1.74 | 0.34 | 0.45 |
| | VISTA3D | 1.72 | 2.71 | 0.45 | 0.69 |
| | SAM-Med3D | 0.30 | 0.02 | 0.08 | 0.00 |
| PET | SegVol | 2.97 | 2.86 | 0.74 | 0.71 |
| | ENSAM (ours) | 2.16 | 1.94 | 0.51 | 0.45 |
| | VISTA3D | 2.39 | 2.10 | 0.61 | 0.54 |
| | SAM-Med3D | 2.13 | 1.82 | 0.53 | 0.46 |
| Ultrasound | SegVol | 1.24 | 1.80 | 0.31 | 0.45 |
| | ENSAM (ours) | 2.10 | 2.41 | 0.55 | 0.62 |
| | VISTA3D | 2.60 | 2.61 | 0.71 | 0.72 |
| | SAM-Med3D | 1.36 | 1.81 | 0.39 | 0.51 |

we do not ensemble predictions or perform augmentations during evaluation, to ensure performance is not artificially inflated in comparison to the other methods. Thus, any observed performance gains should stem from methodological advancements, and not increased compute budget, training data, or test-time tricks.

### 4.3   Qualitative results on validation set

In this section, we provide examples of relatively successful segmentations, as well as interesting failure cases for images in each of the five modalities. For each modality, we compare ENSAM's outputs against the all-data submissions from SAM-Med3D, VISTA3D, and SegVol. Each of which was trained on roughly ten times more annotated volumes than ENSAM.

Figure 5 presents two AbdomenAtlas CT examples: In the first, ENSAM accurately delineates the liver, spleen, and kidneys. In the second, ENSAM oversegments some parts in the left and middle parts of the slice.

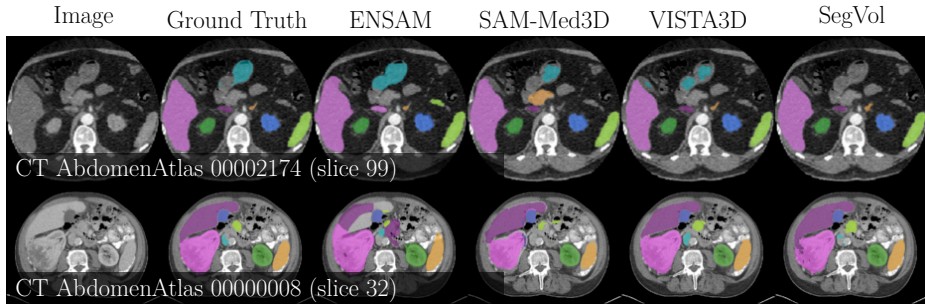

Fig. 5: **Top row:** Shows an example from the AbdomenAtlas dataset where ENSAM successfully segments the liver, spleen, and kidneys. **Bottom row:** Shows an example from the same dataset where ENSAM oversegments some parts.

Figure 6 presents two MRI slices: a slice from the Spider dataset where ENSAM captures the central vertebral bodies but fails at posterior elements; a slice from the TotalSeg dataset illustrating correct localization of large structures but failure on smaller structures.

Figure 7 illustrates PET segmentation: both ENSAM and the baseline models provide similar outputs but are often over- or undersegmented, likely due to intensity clipping during preprocessing.

Figure 8 highlights microscopy challenges: first, a slice from a microscopy volume, dense, small regions where some baseline models oversegment the image. Second, a slice with vessels where all models fail to detect thin vasculature. This might reflect the absence of microscopy data in the coreset when training

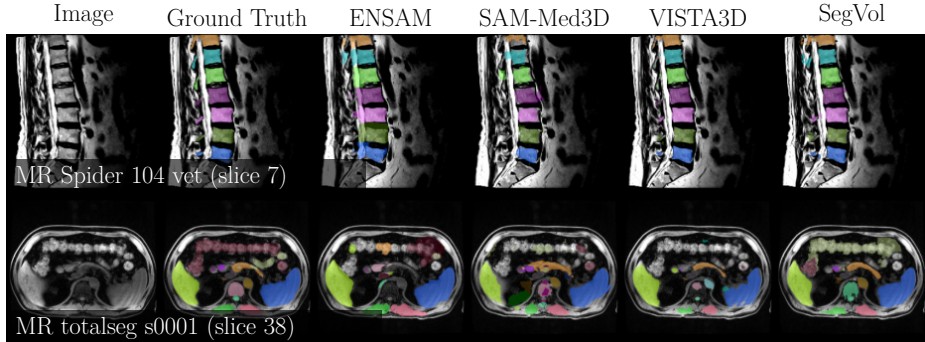

Fig. 6: **Top row:** Shows an example from the MR Spider dataset, where ENSAM successfully segments the body of the vertebra but then fails to segment the posterior. **Bottom row:** shows an example where ENSAM is relatively successful in segmenting the bottom part, whereas all models fail in segmenting the top part of the slice.

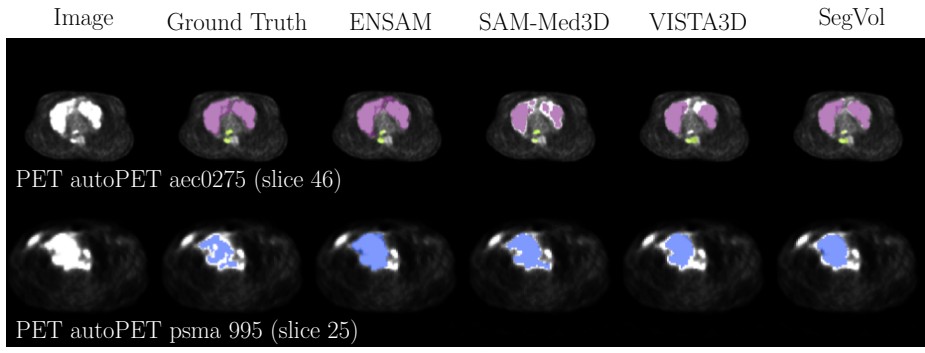

Fig. 7: **Top row:** Shows a PET image where ENSAM successfully segments high-uptake regions. **Bottom row:** Shows a PET image where ENSAM oversegments a high-uptake region. Due to the preprocessing of the PET images, all values in their neighbourhood are maximally bright, possibly making it difficult for ENSAM to distinguish the borders.

ENSAM (see Section 2.5), but also the general ambiguity in segmenting vessels using sparse prompts like single points.

Figure 9 presents two ultrasound frames: The first is a frame from a cardiac 2D+t video where ENSAM outlines the left ventricle and atrium with jagged edges from motion and speckle noise; The second shows a freehand-leg reconstruction slice segmenting three lower-leg muscles with minor boundary artifacts. For this case, SAM-Med3D's inference code crashed, and the other baseline models severely undersegmented all three muscles.

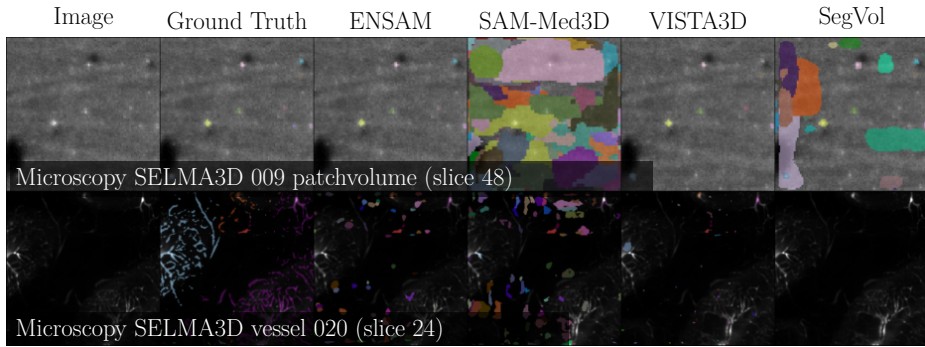

Fig. 8: **Top row:** Shows an example of a microscopy image where all labelled areas are very small, making it easy to obtain high scores for surface-distance-based metrics. It is hard to visually determine how well the model performs on this slice. In this case, SAM-Med3D and SegVol oversegment the volume. **Bottom row:** Shows an example where all models fail to segment the vessels.

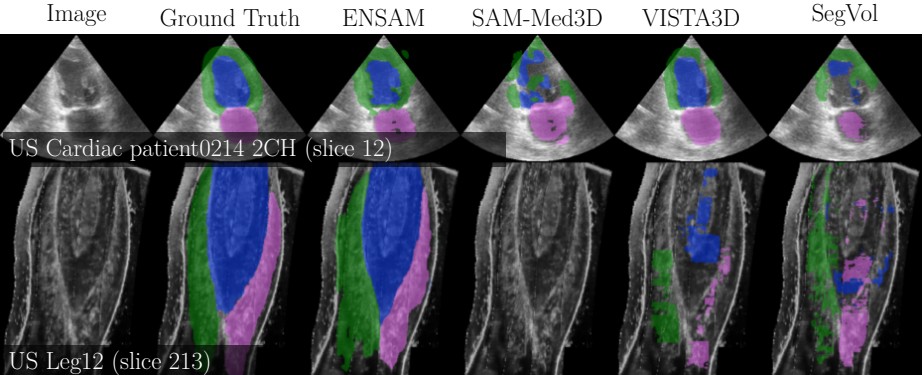

Fig. 9: **Top row:** Shows a frame in a cardiac ultrasound video, where the left atrial walls, blood volume, and left atrium are annotated. The surface of EN-SAM's predictions is not as smooth as the annotations. **Bottom row:** Shows a slice from a 3D volume reconstructed from 2D handheld ultrasound. Three muscles in the lower leg are annotated and segmented relatively successfully by ENSAM. For this case, the inference code provided for SAM-Med3D crashed.

## 4.4   Results on final testing set

Results for the hidden test set were calculated by challenge participants and baseline contributors submitting Docker containers to the challenge organizers, ensuring fair comparison.

**Comparison against baseline models** Qualitative results when compared to previously published baselines are summarized in Table 4. ENSAM outperforms VISTA3D and SAM-MED3D. Across all modalities and metrics, either ENSAM or SegVol performs the best. Notably, ENSAM performs better on the hidden test set as compared to the validation set. This likely reflects the uniform sampling strategy used during training: its broad coverage supports generalization to the diverse hidden test set but is less optimal for the unbalanced validation set.

Table 4: Quantitative evaluation results on the test set for the coreset track. The maximum value for DSC AUC and NSD AUC is 4, while the maximum value for DSC Final and NSD Final is 1.

| Modality | Method | DSC AUC | NSD AUC | DSC Final | NSD Final |
|---|---|---|---|---|---|
| CT | SegVol | 2.27 | **2.16** | 0.57 | 0.54 |
| | ENSAM (ours) | **2.35** | 2.02 | **0.63** | **0.56** |
| | VISTA3D | 2.22 | 1.99 | 0.58 | 0.53 |
| | SAM-Med3D | 2.12 | 1.75 | 0.54 | 0.45 |
| MRI | SegVol | **2.74** | **3.03** | **0.68** | **0.76** |
| | ENSAM (ours) | 2.45 | 2.52 | 0.63 | 0.65 |
| | VISTA3D | 2.41 | 2.51 | 0.63 | 0.67 |
| | SAM-Med3D | 2.12 | 2.12 | 0.54 | 0.54 |
| Microscopy | SegVol | **2.74** | **3.88** | **0.68** | **0.97** |
| | ENSAM (ours) | 2.57 | 3.76 | 0.67 | 0.94 |
| | VISTA3D | 1.84 | 2.50 | 0.47 | 0.63 |
| | SAM-Med3D | 0.31 | 0.03 | 0.08 | 0.01 |
| PET | SegVol | **2.78** | **2.30** | **0.70** | **0.57** |
| | ENSAM (ours) | 2.28 | 1.84 | 0.56 | 0.46 |
| | VISTA3D | 1.76 | 1.28 | 0.45 | 0.33 |
| | SAM-Med3D | 2.31 | 1.76 | 0.58 | 0.45 |
| Ultrasound | SegVol | 0.79 | 1.29 | 0.20 | 0.32 |
| | ENSAM (ours) | **1.79** | **2.04** | **0.47** | **0.55** |
| | VISTA3D | 0.76 | 0.82 | 0.24 | 0.31 |
| | SAM-Med3D | 0.60 | 0.40 | 0.15 | 0.10 |

### 4.5   Unified analysis of validation results

When jointly considering the quantitative and qualitative validation results, several trends emerge. First, the baseline models generally perform strongly on CT and MR, where SegVol and VISTA3D consistently outperform ENSAM in both AUC-based metrics and final DSC/NSD scores (Table 3). This reflects the relative advantage of methods trained on larger or more modality-specific datasets; some of the baselines also include the validation set for pretraining.

Second, it is worth noting that the AUC metrics reported in Table 3 are not directly visible in the qualitative plots, which only illustrate the final segmentation after the last user interaction. In this final step, ENSAM often produces competitive or visually appealing results, particularly in some ultrasound and microscopy cases.

Third, the hidden test set provides additional evidence: ENSAM slightly surpasses all baselines in the final DSC metric after five refinement iterations Table 4. This suggests that ENSAM, although initially trailing, converges to a performance level comparable to state-of-the-art baselines after sufficient user interaction.

Finally, it should be emphasized that the validation set was imbalanced in terms of dataset composition, with many samples drawn from only a few datasets. This imbalance may have biased performance estimates for certain modalities, especially microscopy, and limits the reliability of conclusions drawn solely from validation results.

**Comparison against other challenge participants** Table 5 summarizes the performance of the coreset track participants. Our submission ranked 5th out of 10 teams, placing highest among the participants not leveraging any external pretrained weights.

Table 5: Quantitative evaluation results on the test set for the coreset track. The team names correspond to the names on the official leaderboard. DSC AUC and NSD AUC range from 0 to 4; higher is better.

| Team | Initialization | DSC AUC | NSD AUC | DSC Final | NSD Final |
|---|---|---|---|---|---|
| aim [10] | EfficientTAM | 3.10 | 3.23 | 0.80 | 0.89 |
| norateam [33] | nnInteractive | 2.91 | 2.97 | 0.75 | 0.78 |
| yiooo [43] | VISTA3D & SAM-Med3D | 2.90 | 2.90 | 0.75 | 0.75 |
| lexor [1] | SegVol | 2.50 | 2.57 | 0.63 | 0.64 |
| ahus (ours) | - | 2.40 | 2.27 | 0.63 | 0.60 |
| hanglok [15] | VISTA3D | 2.12 | 2.01 | 0.55 | 0.53 |
| cemrg [35] | - | 1.89 | 1.62 | 0.48 | 0.41 |
| sail [16] | SAM-Med2D | 1.65 | 1.59 | 0.41 | 0.40 |
| dtftech [12] | SegVol | 1.59 | 1.08 | 0.42 | 0.28 |
| owwwen [21] | SAM-Med3D | 0.96 | 0.50 | 0.24 | 0.12 |

### 4.6   Limitation and future work

**Limitations to ENSAM.** There are several limitations to our work, both in the training and inference setup, as well as the model architecture.

First, our model was trained under constrained data and computational budgets. Leveraging the full dataset alongside increased computational resources would likely yield improved performance.

Second, while we conducted targeted ablation studies demonstrating that the Muon optimizer outperforms AdamW and that relative positional encoding is preferable to absolute positional encoding, we did not evaluate the impact of normalized attention compared to the standard attention mechanism. Additionally, due to computational constraints, we did not explore variations in model size. It is therefore unlikely that the current architecture is optimal; for instance, increasing the depth of the U-Net may lead to better results.

Third, as noted by Isensee et al. [9], 2D bounding boxes can be preferable to 3D ones even for 3D segmentation tasks, and click-based prompts may convey less information compared to other prompt types. However, as the evaluation protocol for this challenge relies on 3D boxes and clicks, the current version of ENSAM supports only these prompt types.

Fourth, we have not conducted latency evaluations or user studies. To robustly assess the practical utility of interactive segmentation models, simulated user interactions alone are insufficient; real user studies are essential.

**Limitations in evaluation pipeline** While the test set contains unpublished medical images and annotations providing a fair comparison between models, the validation set includes mostly CT and MR images. In the PET modality, a single dataset is included, and for US, two datasets are used. One of the US datasets comprises 2D+t echocardiographic videos. The microscopy modality includes just eight volumes, several of which contain only a small fraction of labelled voxels. As a result, quantitative conclusions regarding model performance on US, PET, and Microscopy should be interpreted with caution due to limited sample diversity and volume.

In addition, the CT and MR subsets of the validation set are imbalanced in terms of the number of samples taken from each dataset. This imbalance may bias the evaluation and obscure insights into how well the model generalizes across datasets within a modality. A more robust assessment, especially for modalities like CT, could be achieved through a more uniform sampling strategy that considers factors such as the number of labeled instances per dataset.

Lastly, the simulation of user input could be improved, aligning better with the use case of the model. Currently, $N$ interactions are provided between each refinement step, with $N$ being equal to the number of objects of interest in the image. In reality, the user would probably want an updated segmentation after each interaction.

**Future directions** The field is rapidly evolving, and future work should focus on improving multiple areas. Below, we outline promising directions.

**User input integration via attention:** The comparative effectiveness of attention based methods versus dense feature maps to incorporate user input warrants further investigation. In particular, prompt types such as scribbles and lassos have not yet been explored in the context of attention mechanisms in 3D medical images.

**Handling anisotropic spacing and physical coordinates:** The impact of anisotropic voxel spacing on position encoding remains an open question. In this work, voxel spacing is disregarded, but future studies could examine whether representing coordinates in physical units improves model performance. A related challenge is the incorporation of spatiotemporal data in training and evaluation.

**Handling multiple objects:** Current methods, including ENSAM and all baseline models of the challenge, treat each object instance in parallel during training and inference. Incorporating interactions between instances could likely reduce the amount of required user input and improve inference efficiency. For example, a foreground click for one instance could be interpreted as a background click for others.

## 5   Conclusion

In this paper, we presented ENSAM, an efficient, promptable model for universal medical image segmentation. With compute restraint and training on a coreset of the challenge data, we achieved a DSC AUC of 2.404, NSD AUC of 2.266, final DSC of 0.627, and final NSD of 0.597. Our results demonstrate that the combination of relative positional encoding and the Muon optimizer significantly improves both model performance and training efficiency. Furthermore, enabling the model to handle variable-shape inputs is critical for reducing computational overhead, particularly VRAM usage, and facilitates inference at resolutions closer to the native input scale.

**Acknowledgements**  We thank all the data owners for making the medical images publicly available and CodaLab [41] for hosting the challenge platform. We also thank Akershus University Hospital for the funding that made this study possible.

**Disclosure of Interests**  The authors have no competing interests to declare that are relevant to the content of this article.

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
