# OpenReview forum: "ENSAM: an efficient foundation model for interactive segmentation of 3D medical images"
_thecvf.com/CVPR/2025/Workshop/MedSegFM — CVPR 2025 Workshop MedSegFM Submission_

### Official Review · Reviewer_6gWS · 2025-09-08
**This paper proposes a lightweight method for universal 3D medical segmentation. The method is based on SegResNet model, and by introducing some recent techniques, ENSAM can achieve competitive performance with the baseline models, underscoring its effectiveness.**

**Rating:** 9
**Confidence:** 4

**Review:**

The overall paper is well-structured, and all key points of ENSAM are clearly illustrated. The model builds on the SegResNet and introduces some of the latest techniques, including relative positional encoding, normalized attention, and the Muon optimizer. Furthermore, some changes are made to ensure the effective interaction between image embeddings and prompts; also, previous segmentation results are considered as well.

For the weakness of paper, some ablation studies are missing, like the validation of the normalized attention part; it is better to enrich it further in the later version. Although ENSAM can achieve competitive performance without using any pretrained weights, it is still better to consider introducing it, as it can have some potential benefits for the current model to achieve better performance and faster convergence.

---

> ### Author Response · Authors · 2025-09-29
> **Response to reviewer**
>
> We thank the reviewer for the kind comments.
>
> Indeed, ablation studies on normalized attention are not performed; it is listed as a limitation of the current work.
>
> We agree that using pretrained weights would result in better performance and faster convergence. However, we have left it out for two main reasons: (1) baselines might have been trained on parts of the official validation set, which would artificially inflate the validation results, and (2) we wish to demonstrate that it is possible to train a foundation model from scratch with competitive performance under compute and data constraints.

---

### Official Review · Reviewer_Jkmv · 2025-09-10
**ENSAM is a thoughtfully designed approach, but stronger empirical validation and deeper analysis are needed to support its advantages.**

**Rating:** 7
**Confidence:** 4

**Review:**

Overall: This manuscript proposes an Equivariant and Normalized SAM (ENSAM) model to achieve strong performance under limited data and computational budgets. The method integrates several advanced components — including Lie Rotational Positional Encoding, Normalized Attention, and the Muon optimizer — and is thoughtfully designed for the interactive segmentation scenario emphasized by the workshop. The experimental results show that ENSAM achieves comparable performance with baselines. Furthermore, the manuscript is well-structured and clearly written. However, several critical issues require attention:

Detailed Comments:
1. Table 3 presents comparative results across multiple modalities, yet the proposed ENSAM method shows limited or no improvement over existing baselines. The authors should provide a detailed analysis and justification for why the method underperforms quantitatively.
2. There is inconsistency between Quantitative and Qualitative results. It is puzzling that while quantitative metrics (Table 3) show marginal or inferior performance, the qualitative results (Figs. 8 & 9) appear visually superior. The authors must reconcile this discrepancy — for instance, by discussing which aspects of segmentation quality are captured visually but not reflected in DSC/NSD metrics, or whether evaluation metrics are misaligned with perceptual quality.
3. The manuscript requires careful proofreading. For example, on page 12, the sentence “For all experiments” appears incomplete. Similar minor grammatical or formatting issues should be corrected to ensure professional presentation.
4. The experimental section reports results but lacks critical analysis. Beyond presenting tables and figures, the authors should interpret what the results imply.

---

> ### Author Response · Authors · 2025-09-29
> **Response to reviewer**
>
> We thank the reviewer for the constructive feedback. We have addressed the main concerns as follows:
>
> Unified analysis and critical discussion: We added Section 4.5 to reconcile quantitative and qualitative results. This clarifies that AUC metrics (Table 3) are not visible in qualitative plots, which only show final predictions where ENSAM often performs competitively. On the hidden test set, ENSAM slightly surpasses all baselines in final DSC, indicating convergence after five iterations, also partially explaining the qualitative performance. As for Figure 8, we note that it is difficult to determine performance as all annotated volumes are quite small.
>
> Proofreading:  We have identified and fixed typographical errors.

---

### Official Review · Reviewer_KZHP · 2025-09-29
**ENSAM**

**Rating:** 7
**Confidence:** 4

**Review:**

This paper proposes a lightweight and promptable model, termed ENSAM, for universal 3D medical segmentation. ENSAM adopts an end-to-end architecture with dual encoders and a mask decoder, and further incorporates latent cross-attention, relative positional encoding, normalized attention, and the Muon optimizer to enhance performance. Overall, the manuscript is clearly written and well structured, and the proposed ENSAM demonstrates both efficiency and effectiveness.
Nevertheless, I have two concerns that should be addressed to strengthen the paper:
1. In Table 3, ENSAM does not achieve the best performance across the five evaluated modalities. The authors should provide a more detailed discussion clarifying the advantages of ENSAM over competing methods such as SegVol and VISTA3D.
2. The results section would benefit from more in-depth analysis. Moreover, more detailed ablation studies would significantly enhance the technical rigor of the work.

---

> ### Author Response · Authors · 2025-10-08
> **Response to reviewer**
>
> We thank the reviewer for the comments, and have done the following to address the feedback:
>
> We have added a section to reconcile the quantitative and qualitative results. Further, while SegVol and VISTA3D outperform ENSAM on the public validation set (Table 3), it is not the case in the hidden test set (Table 4), possibly indicating that the baseline models were trained on the validation set, as part of their initial pretraining.

---

### Author Rebuttal · Authors · 2025-10-31

We thank the reviewers for providing useful feedback that has helped us strengthen the paper. We have uploaded the revised paper and responded separately to each reviewer.

---

### Decision · Program_Chairs · 2025-11-12

Accept